# Molecular architecture of thylakoid membranes within intact spinach chloroplasts

**Wojciech Wietrzynski[1]\*[†], Lorenz Lamm[1,2][†], William HJ Wood[3], Matina-Jasemi Loukeri[4], Lorna Malone[3][‡], Tingying Peng[2], Matthew P Johnson[3]\*, Benjamin D Engel[1]\***

[1]Biozentrum, University of Basel, Basel, Switzerland; [2]Helmholtz AI, Helmholtz Zentrum München, Neuherberg, Germany; [3]Plants, Photosynthesis and Soil, School of Biosciences, University of Sheffield, Sheffield, United Kingdom; [4]Department of Structural Cell Biology, Max Planck Institute of Biochemistry, Martinsried, Germany

**\*For correspondence:**
wojciech.wietrzynski@unibas.ch (WW);
matt.johnson@sheffield.ac.uk (MPJ);
ben.engel@unibas.ch (BDE)

[†]These authors contributed equally to this work

**Present address:** [‡]Electron Bio-imaging Centre, Diamond Light Source, Didcot, United Kingdom

## eLife Assessment

The macromolecular organization of photosynthetic complexes within the thylakoids of higher plant chloroplasts has been a topic of significant debate. Using in situ cryo-electron tomography, this study reveals the native thylakoid architecture of spinach thylakoid membranes with single-molecule precision. The experimental methods are unique and **compelling**, providing **important** information for understanding the structural features that impact photosynthetic regulation in vascular plants and addressing several long-standing questions about the organization and regulation of photosynthesis.

**Abstract** Thylakoid membranes coordinate the light reactions of photosynthesis across multiple scales, coupling the architecture of an elaborate membrane network to the spatial organization of individual protein complexes embedded within this network. Previously, we used in situ cryo-electron tomography (cryo-ET) to reveal the native thylakoid architecture of the green alga *Chlamydomonas reinhardtii* (Engel et al., 2015) and then map the molecular organization of these thylakoids with single-molecule precision (Wietrzynski et al., 2020). However, it remains to be shown how generalizable this green algal blueprint is to the thylakoids of vascular plants, which possess distinct membrane architecture subdivided into grana stacks interconnected by non-stacked stromal lamellae. Here, we continue our cryo-ET investigation to reveal the molecular architecture of thylakoids within intact chloroplasts isolated from spinach (*Spinacia oleracea*). We visualize the fine ultrastructural details of grana membranes, as well as interactions between thylakoids and plastoglobules. We apply AI-based computational approaches (Lamm et al., 2024) to quantify the organization of photosynthetic complexes within the plane of the thylakoid membrane and across adjacent stacked membranes. Our analysis reveals that the molecular organization of thylakoid membranes in vascular plants and green algae is strikingly similar. We find that PSII organization is non-crystalline and has uniform concentration both within the membrane plane and across stacked grana membranes. Similar to *C. reinhardtii*, we observe strict lateral heterogeneity of PSII and PSI at the boundary between appressed and non-appressed thylakoid domains, with no evidence for a distinct grana margin region where these complexes have been proposed to intermix. Based on these measurements, we support a simple two-domain model for the molecular organization of thylakoid membranes in both green algae and plants.

## Introduction

Biological membranes and the proteins embedded within them are interwoven components of a dynamic multiscale system. Thylakoid membranes present one of the best examples of such a system, with interconnected relationships between membrane architecture, molecular organization, and physiological function (*Wietrzynski et al., 2023*; *Ostermeier et al., 2024*). Therefore, to fully understand the light-dependent reactions of photosynthesis, it is crucial to describe thylakoid architecture with high precision in its most native state.

Thylakoids are the basic unit of oxygenic photosynthesis (with the exception of *Gloeobacteria*, which completely lack them *Rippka et al., 1974*; *Rahmatpour et al., 2021*). They form a variety of architectures in cyanobacteria and within the chloroplasts of plants and algae (*Perez-Boerema et al., 2024*). In vascular plants, thylakoids assemble into one of the most intricate membrane networks in living organisms: one continuous compartment that is folded and flattened to form interconnected cisternae (*Austin and Staehelin, 2011*; *Nevo et al., 2012*; *Gounaris et al., 1986*). Thylakoid shape and packing within plant chloroplasts are optimized to maximize the surface area for light absorption and the coupled electron and proton transfer reactions conducted by photosynthetic electron transport chain complexes. The thylakoid network is divided into domains of appressed, stacked membranes (called grana) and non-appressed, free thylakoids (called stromal lamellae) that interconnect the grana. This architecture is crucial for the spatial separation of photosystem I (PSI) and photosystem II (PSII) and their associated light harvesting antenna complexes, LHCI and LHCII, thereby preventing the 'spillover' (and thus loss) of excitation energy from the short wavelength trap of PSII (680 nm) to the longer wavelength trap of PSI (700 nm) (*Kitajima and Butler, 1975*; *Murata, 1969*). The existence of two thylakoid domains and their dynamics has been shown to underpin a variety of regulatory mechanisms that control linear electron transfer (LET), cyclic electron transfer (CET) (*Garty et al., 2024*; *Wood et al., 2018*), the PSII repair cycle (*Puthiyaveetil et al., 2014*), non-photochemical quenching (NPQ) (*Johnson et al., 2011*), and the balance of excitation energy distributed to PSI and PSII (*Kyle et al., 1983*; *Wood et al., 2018*). Despite over half a century of efforts using a multitude of techniques, including freeze-fracture electron microscopy (EM), negative-stain EM, atomic force microscopy, and fluorescence microscopy, our understanding of the organizational principles of thylakoid networks remains limited by resolution and the requirement for intactness of the system (*Staehelin, 2003*; *Staehelin and Paolillo, 2020*; *Shimoni et al., 2005*; *Bos et al., 2023*; *Mustárdy et al., 2008*; *Ruban and Johnson, 2015*). Indeed, numerous controversies remain, including the organization of the grana margins and grana end membranes (*Albertsson, 2001*), the architecture of the connections between grana and stromal lamellae (*Austin and Staehelin, 2011*; *Shimoni et al., 2005*; *Bussi et al., 2019*; *Daum et al., 2010*), and the exact distribution of protein complexes such as ATP synthase (ATPsyn) and cytochrome $b_6f$ (cyt$b_6f$) (*Johnson et al., 2014*; *Kirchhoff et al., 2017*).

A decade ago, we combined cryo-focused ion beam (FIB) milling (*Schaffer et al., 2017*; *Lam and Villa, 2021*) with cryo-electron tomography (cryo-ET) (*Nogales and Mahamid, 2024*; *Young and Villa, 2023*) to provide a detailed view of thylakoid architecture in situ, inside the chloroplasts of *Chlamydomonas reinhardtii* cells (*Engel et al., 2015*). We then built upon this by leveraging advances in direct electron detector cameras and the contrast-enhancing Volta phase plate to understand the distribution of photosynthetic complexes within the native thylakoid network of *C. reinhardtii* cells (*Wietrzynski et al., 2020*). Now, we apply in situ cryo-ET to the more challenging system of plant chloroplasts, utilizing an artificial intelligence (AI)-assisted approach (*Lamm et al., 2024*; *Lamm et al., 2022*) to identify individual photosynthetic complexes and quantify their organization within networks of grana and stromal lamellae membranes.

## Results and discussion

We isolated chloroplasts from six-week-old spinach plants, then plunged them onto EM grids, thinned them by cryo-FIB milling, and finally imaged them by cryo-ET with defocus contrast (no Volta phase plate). To enhance the contrast of the defocus imaging, we used AI-based denoising algorithms (*Buchholz et al., 2019*; *Wiedemann and Heckel, 2024*). The resulting tomograms provide a glimpse into the near-native organization of the spinach plastid (*Figure 1A*, *Figure 1—figure supplement 1*, *Figure 1—video 1*). The double-membrane envelope encloses a protein-dense stroma (*Figure 1B*, segmentations: envelope in blue, stroma proteins in gray. Also note the high concentration of Rubisco

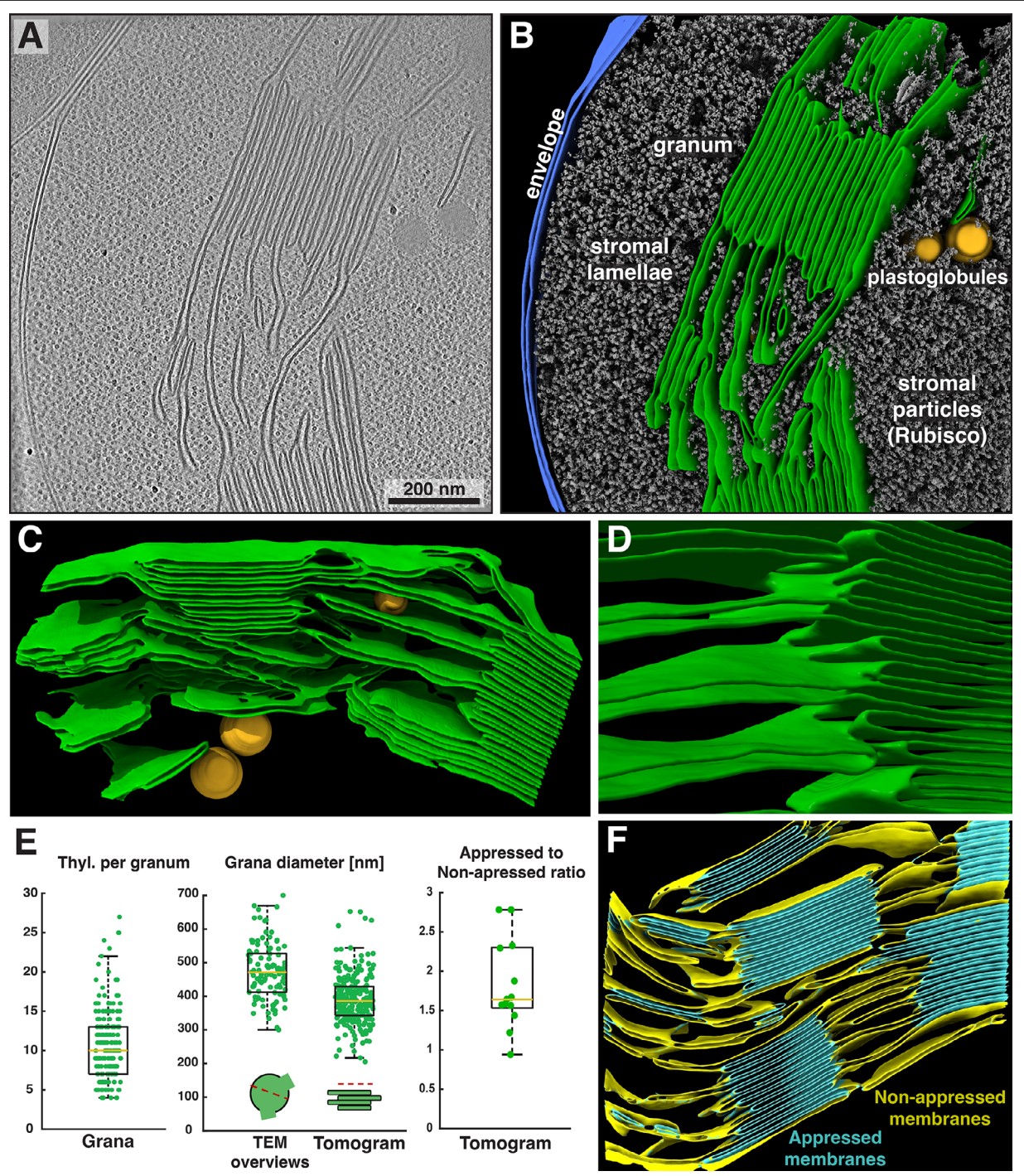

**Figure 1.** Overview of intact spinach chloroplasts visualized by cryo-electron tomography (cryo-ET). (**A**) Tomographic slice through an intact spinach chloroplast. (**B**) Corresponding 3D segmentation of the volume. Characteristic elements are labeled. Thylakoid membranes (green) are organized into stacks of appressed thylakoids (grana) and individual thylakoids connecting the grana (stromal lamellae). Note the intact chloroplast envelope (blue) and single thylakoid surrounded by two plastoglobules (orange). Stroma particles (gray) are a mix of different complexes but predominantly Rubisco. (**C**) The same thylakoid network as in B, here shown from the opposite side. Note the inclined stromal lamellae spiraling around the granum on the left and a smaller plastoglobule sandwiched between the membranes. (**D**) Close-up on the margin of a granum (from a different tomogram) showing the individual thylakoids transitioning into inclined stromal lamellae. In the first transition at the top of the image, note the hole in the thylakoid where the stromal lamella meets the granum. (**E**) Box plots showing the number of thylakoids per granum (left), grana diameter measured in TEM overviews and in tomograms (middle), and the ratio of appressed to non-appressed membrane surfaces measured in selected tomograms. Box: 75 percentile, yellow lines: mean, whiskers: 95 percentile. (**F**) Representative segmentation and classification of appressed (teal) and non-appressed (yellow) thylakoid

*Figure 1 continued on next page*

*Figure 1 continued*

membranes (quantification shown in E). See *Figure 1—video 1* for an additional example of a chloroplast tomogram and segmentation of the thylakoid network.

The online version of this article includes the following video and figure supplement(s) for figure 1:

**Figure supplement 1.** Sample preparation, tomogram selection on the focused ion beam (FIB)-milled lamellae, and types of tomograms collected.

**Figure supplement 2.** Thylakoid-chloroplast envelope contacts.

**Figure supplement 3.** Ultrastructure of grana sides and transition to stromal lamellae.

**Figure 1—video 1.** Overview of a chloroplast tomogram, slicing through the tomographic volume and then revealing segmentations of the thylakoid and chloroplast envelope membranes.

https://elifesciences.org/articles/105496/figures#fig1video1

throughout the chloroplast, e.g., *Figures 2A and 3*; orange arrowheads point to Rubisco particles in *Figure 3B*). Suspended within the stroma, the thylakoid network is the main architectural feature of the chloroplast (*Figure 1A–C*, segmented in green). The thylakoids form a discrete network that is discontinuous from the surrounding chloroplast envelope. However, we did observe rare contact sites between the thylakoid and envelope membranes (*Figure 1—figure supplement 2*), consistent with previous cryo-ET observations in *C. reinhardtii* (*Engel et al., 2015*; *Gupta et al., 2021*). Spherical plastoglobules (*Shanmugabalaji, 2022*) decorated in putative protein densities were often seen interacting with thylakoid surfaces (*Figures 1B, C and 2*, segmented in orange). We noticed holes through the thylakoid sheets precisely at the plastoglobule contact sites (*Figure 2D–E*). At those sites, plastoglobules attach to thylakoids along an extended stretch of high curvature thylakoid membrane, as opposed to making a single point contact to a flat thylakoid sheet (*Figure 2F*). This may help facilitate exchange between the two compartments.

Segmentation of the membranes highlights the nature of the folded thylakoid network (*Figure 1C and D*, *Figure 1—figure supplement 3*). The grana form well-defined stacks of appressed membranes, which, when viewed from the top, were revealed to be irregular cylinders with wavy edges (*Figure 3A*, *Figure 3—video 1*). Sometimes, thylakoid layers change length and extend laterally beyond the stack, breaking the grana's vertical order (e.g. *Figure 1—figure supplement 3A*). Distances between grana differ, and in some instances, they almost merge with one another (Figure 5A and B, *Figure 1—figure supplement 3A*, *Figure 1—video 1*). In our dataset, the number of thylakoids per grana stack varied from 4 to 27, with a mean of 10 (*Figure 1E* left). Every granum connects with the stromal lamellae membranes that spiral around each stack, in accordance with the helical staircase model (*Figures 1C, D and 3A*; *Austin and Staehelin, 2011*; *Mustárdy et al., 2008*). Stromal lamellae extend from the stacks at variable angles, and after some distance, join other grana or rarely dead-end in the stroma (*Figure 1C and D*, *Figure 1—figure supplement 3A*). Our cryo-ET volumes reveal the fine structural details of near-native membrane bilayers at high resolution (e.g. *Figure 4* left), allowing us to precisely quantify the morphometric parameters of thylakoids with sub-nanometer precision. We measured a membrane thickness of 5.1±0. 3 nm, a stromal gap of 3.2±0. 3 nm, a luminal thickness of 10.8±2.0 nm, and a total thylakoid thickness (including two membranes plus the enclosed lumen) of 21.1±1.8 nm (*Figure 4*) (for comparison see *Engel et al., 2015*; *Wietrzynski et al., 2020*; *Kirchhoff et al., 2017*; *Kirchhoff et al., 2011*; *Li et al., 2020*).

It is important to note that individual tomograms represent only a small slice of the chloroplast volume (approximately 1200 × 1200 × 120 nm). This makes sampling of the larger micrometer-scale features less statistically relevant (see *Mazur et al., 2021* and references within for alternative approaches). For example, grana diameter measured in our cryo-ET volumes is inherently underestimated due to FIB milling yielding only a narrow slice through the grana stack (*Figure 1E* middle). We, therefore, complemented these measurements with TEM overviews of FIB-milled lamellae, where many top views of grana were visible (*Figure 1—figure supplement 1B*), facilitating more accurate diameter measurements (*Figure 1E* middle). Similarly, we are able to report the exact ratio between appressed and non-appressed membrane surface areas in our dataset (*Figure 1E* right, *1F*), but we are aware that this ratio may not be representative of the entire chloroplast volume (see Limitations and future perspectives).

Thylakoid membranes are populated primarily by the photosynthetic complexes. Their distribution follows the principle of lateral heterogeneity (*Andersson and Anderson, 1980*): PSII

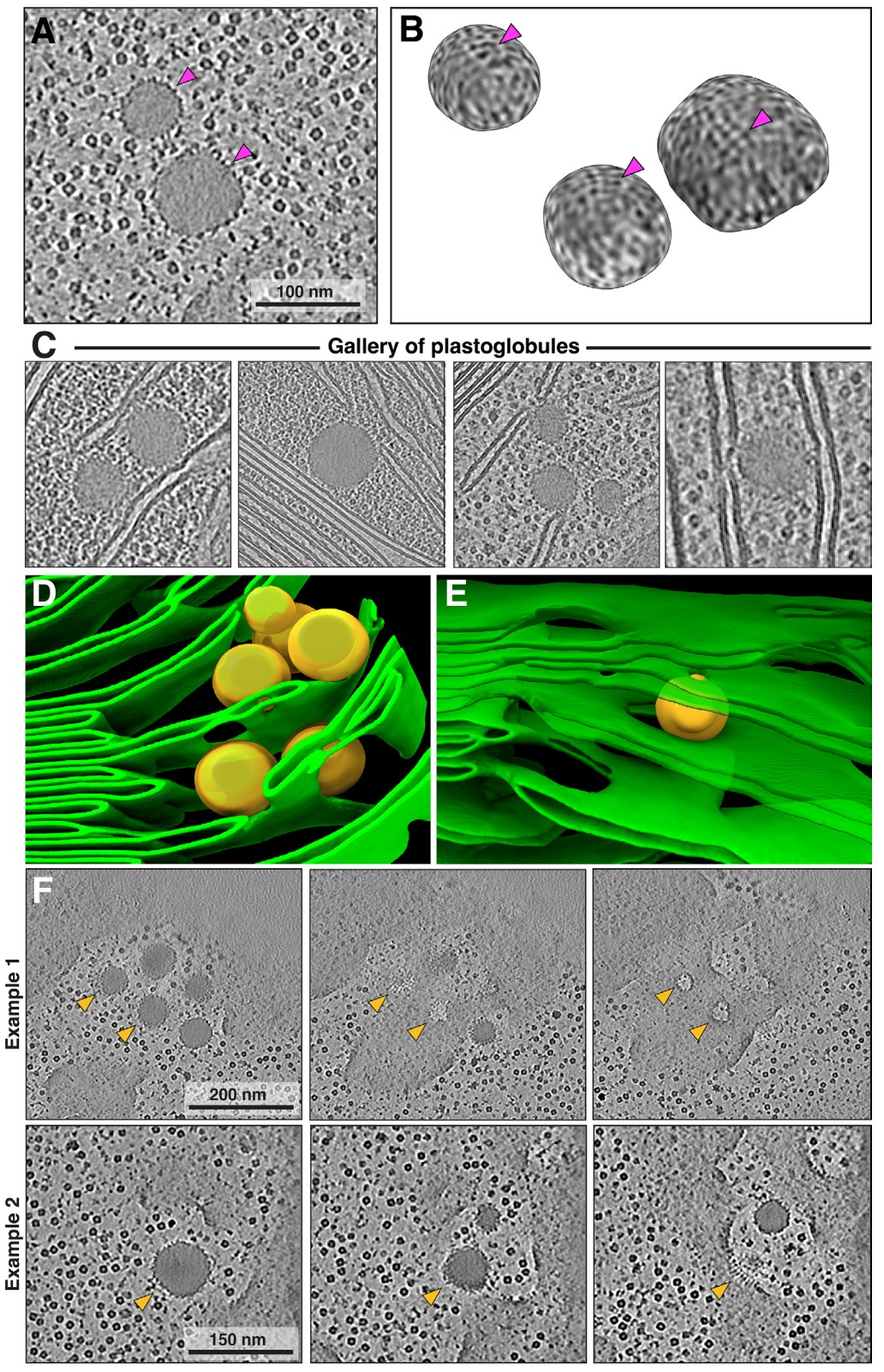

**Figure 2.** Fine details of plastoglobules interacting with thylakoids. (**A**) Tomographic slice showing two plastoglobules: dense, spherical droplets that have surfaces covered with protein densities. (**B**) 3D render of the surfaces of three plastoglobules. Note the ordered, small densities. Magenta arrows in A and B indicate the protein coat. (**C**) Gallery of tomographic slices showing plastoglobules in close proximity of thylakoids and

*Figure 2 continued on next page*

*Figure 2 continued*

the chloroplast envelope (second panel in the gallery of plastoglobules). (**D, E**) Zoom in on a segmentation of thylakoids with visible fenestrations in stromal lamellae. These holes may be caused by interactions with adjacent plastoglobules. (**F**) Two examples of plastoglobules potentially interacting with thylakoids. Each example shows three consecutive tomographic z-slices. Note the protein-membrane interactions in example 2, with an array of small protein densities (potentially coming from the plastoglobule coat) mediating the contact between compartments. Orange arrowheads indicate positions of selected plastoglobules and their interactions with thylakoid membranes.

complexes are localized to appressed membranes, whereas PSI and ATPsyn complexes reside in the non-appressed regions (*Miller and Staehelin, 1976*; *Armond et al., 1977*; *Vallon et al., 1986*; *Armond and Arntzen, 1977*). The size of the stroma-exposed domains of PSI and ATPsyn (~3.5 nm and ~12 nm, respectively) prevents them from entering the narrow stromal gap between appressed thylakoids (~3.2 nm, *Figure 4*). We were able to visualize particle distributions in appressed and non-appressed regions by projecting the tomogram volumes onto both the stromal and luminal surfaces of segmented membranes (*Figure 5A*, *Figure 5—figure supplement 1*) to generate 'membranograms' (*Figures 5C–E and 6A*). Similar to *C. reinhardtii* (*Wietrzynski et al., 2020*) and as observed in other systems (*Daum et al., 2010*; *Levitan et al., 2019*; *Li et al., 2021*), large densities corresponding to the protruding oxygen-evolving complexes (OEC) of PSII are visible on the luminal side of the appressed membranes (e.g. *Figures 3 and 5C–E*, *Figure 3—figure supplement 1D*). These OEC densities mark the center positions of large membrane-embedded supercomplexes of PSII bound to trimers of LHCII (*Perez-Boerema et al., 2024*; *Croce and van Amerongen, 2020*; *Grieco et al., 2015*), which do not extend from the membrane and thus are not visible in the membranograms. Interspersed between the PSII densities, we observe smaller dimeric particles that are consistent with the size of the lumen-exposed domains of cyt$b_6f$, confirming the long-debated presence of this complex in appressed membranes of vascular plant thylakoids (e.g. *Pribil et al., 2014*). In contrast, the stromal surface of the appressed membranes is generally smooth and featureless, consistent with the exclusion of PSI and ATPsyn, as well as the lack of large protrusions on the stromal faces of PSII, LHCII, or cyt$b_6f$. The PSII complexes abruptly disappear from the luminal face at the junction between appressed and non-appressed thylakoids (teal-to-yellow transitions in *Figure 5C and E*; *Figure 5—figure supplement 1A*). Instead, both the luminal and stromal surfaces of the non-appressed thylakoids are covered by a high number of smaller densities (*Figure 5C*, *Figure 5—figure supplement 1*). The identity of these densities is difficult to assign with confidence, but a majority of them likely correspond to PSI (on the stromal side) and cyt$b_6f$ (on the luminal side) (see diagram in *Figure 5F*).

Biochemical isolation experiments often subdivide the non-appressed membranes into stromal lamellae, grana margins, and grana end membranes on the basis of detergent solubilization and mechanical fragmentation, and have reported different protein compositions for the regions (for references see e.g. *Albertsson, 2001*; *Koochak et al., 2019*). In contrast, we observe little difference in the appearance and distribution of the densities on non-appressed membranes irrespective of whether they are immediately adjacent to grana (margins), extend away from them (stromal lamellae), or cap the grana stacks (grana end membranes) (*Figure 5C and E*; *Figure 5—figure supplement 1*). Rather, we observe a simpler two-domain organization of photosynthetic complexes segregated into appressed and non-appressed regions, which corresponds to the original lateral heterogeneity model put forward by *Andersson and Anderson, 1980*; *Anderson and Andersson, 1982*; *Anderson, 1981*. It should be noted that the edges (tips) of the appressed membranes are difficult to visualize with membranograms, although their highly curved nature makes it unlikely that they could accommodate large photosynthetic complexes (see zoomed-in views of thylakoid tips in *Figures 1D and 4*, *Figure 1—figure supplement 3B–D*).

Using AI-assisted approaches (*Lamm et al., 2024*; *Lamm et al., 2022*) confirmed by manual inspection, we detected essentially all particles populating appressed spinach membranes ([All$_{Spin}$]=2160 particles/μm$^2$). Dimeric PSII complexes were clearly identified in the tomograms ([PSII$_{Spin}$]=1415 particles/μm$^2$), enabling us to assign their center positions and orientations (*Figure 6A and C*; *Lamm et al., 2022*). In the best quality tomograms, we were also able to manually assign dimeric cyt$b_6f$ particles in the appressed regions ([cyt$b_6f_{Spin}$]=446 particles/μm$^2$) (*Figure 6A*). Note that the total particle

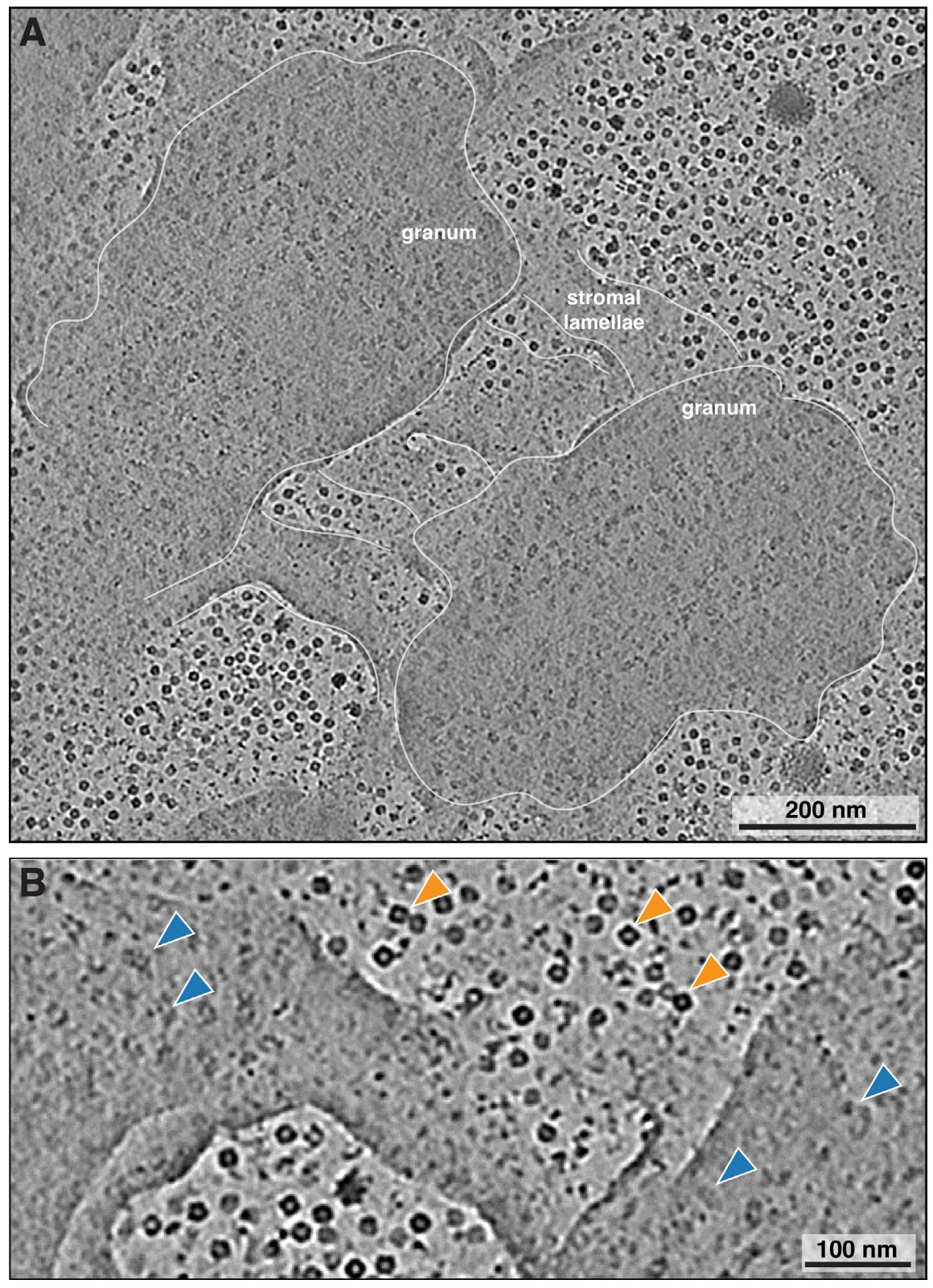

**Figure 3.** Top view of the interconnections between two grana. (**A**) Averaged image of 40 tomographic slices (56 nm total averaged volume) from a tomogram denoised with the DeepDeWedge neural network (**Wiedemann and Heckel, 2024**) (high contrast, missing wedge inpainting), showing top views of two neighboring grana. Note the irregular shape of the grana, with wavy edges (traced with white line). Three stromal lamellae connect the two grana stacks. PSII particles are visible in the stacks. (**B**) Close-up of a region from A, showing a stromal lamella bridging two grana. Blue arrowheads: PSII

*Figure 3 continued on next page*

*Figure 3 continued*

particles in grana membranes; orange arrowheads: Rubisco particles in the stroma. See *Figure 3—video 1* for tomographic slices through the entire volume, highlighting the dense membrane organization of the chloroplast.

The online version of this article includes the following video and figure supplement(s) for figure 3:

**Figure supplement 1.** Top view of a granum.

**Figure 3—video 1.** Slices through the tomogram shown in *Figure 3*, visualizing the chloroplast stroma and top views of the thylakoid network, with stromal lamellae connecting the grana.

https://elifesciences.org/articles/105496/figures#fig3video1

concentration, but not the PSII and cyt$b_6f$ subclasses, includes unknown densities as well as clipped particles at the edges of membrane patches selected for analysis.

The PSII concentration and mean center-to-center distance between nearest neighbor complexes (PSII-NN$_{Spin}$=21.2 ± 3.1 nm) (*Figure 6B*) is similar to what was reported in *Arabidopsis thaliana* by freeze fracture EM (*Goral et al., 2012*). In contrast, our previous cryo-ET measurements of *C. reinhardtii* showed a slightly lower PSII concentration and correspondingly longer nearest neighbor distances within appressed thylakoid regions ([PSII$_{Chlamy}$]=1122 PSII/µm$^2$, PSII-NN$_{Chlamy}$=24.4 ± 5.2 nm) (*Wietrzynski et al., 2020*). This discrepancy is in agreement with the relative sizes of the PSII-light harvesting complex II (PSII-LHCII) supercomplexes isolated from the respective species, with C$_2$S$_2$M$_2$L$_2$ supercomplexes from *C. reinhardtii* having a larger footprint in the membrane than C$_2$S$_2$M$_2$ supercomplexes from vascular plants (*Sheng et al., 2019*; *Wei et al., 2016*; *Graça et al., 2021*). The concentration of PSII complexes is consistent between membranes (*Figure 6B*) and is independent of distance to the edge of the granum and thylakoid position in the stack (*Figure 6E* shows lengths of thylakoids analyzed in *Figure 6D*; we sampled membranes with variable lengths to depict a wide range of thylakoid sizes).

Using the observable orientations of OEC densities within the membrane plane, we placed footprints into the segmented membranes corresponding to structures of the C$_2$ dimeric PSII core complex, the C$_2$S$_2$ PSII-LHCII supercomplex (isolated from spinach; *Wei et al., 2016*, PDB: 3JCU), and the C$_2$S$_2$M$_2$ PSII-LHCII supercomplex (isolated from *A. thaliana*; *Graça et al., 2021*, PDB: 7OUI). When considering the PSII cores alone (C$_2$), only rare events of complexes touching were observed (*Figure 7A* top row, *7D*). Placing in the C$_2$S$_2$ supercomplexes resulted in a small in-plane clash between

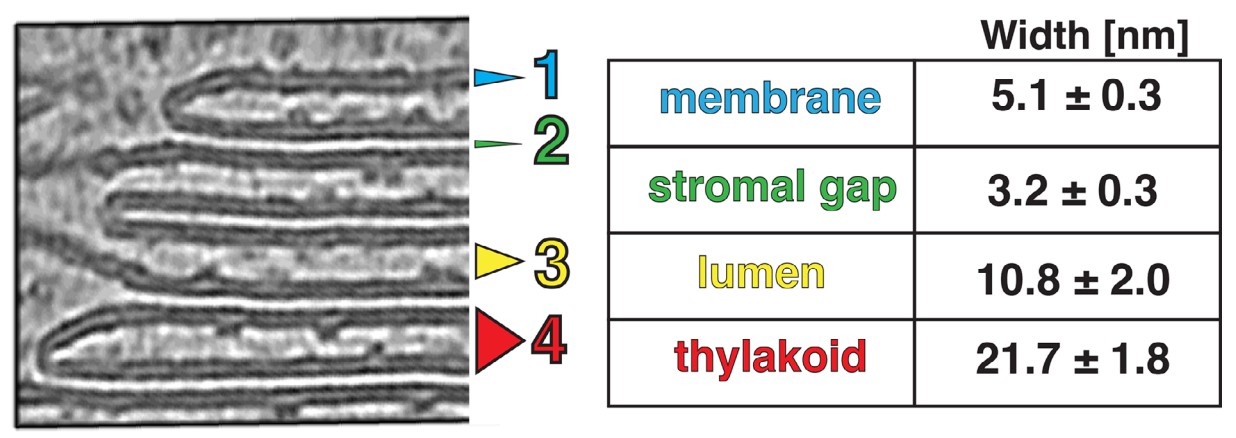

**Figure 4.** Thylakoid spacing and thickness measurements. Left: a zoom into thylakoids at the granum edge, showing the level of details observed in defocus cryo-tomograms denoised with cryo-CARE. Notice how the thylakoid tips stick to the appressed membrane, making them non-symmetrical. Right: quantification of thylakoid morphometric parameters. Membrane (1, blue) is the mean thickness of the lipid bilayer; stromal gap (2, green) is the mean distance between stacked thylakoids; lumen (3, yellow) is the mean thylakoid lumen width; thylakoid (4, red) is mean thickness of the thylakoid including two membrane bilayers plus the enclosed lumen (measured separately from the individual measurements of those features). Errors: SEM. Note the larger SEM values for lumen and thylakoid. This reflects the lumen width variability between rigid appressed thylakoids of the granum and more labile stromal lamellae. *Figure 4—figure supplement 1* details the measurement procedure.

The online version of this article includes the following figure supplement(s) for figure 4:

**Figure supplement 1.** Methodology for measuring thylakoid widths from tomograms.

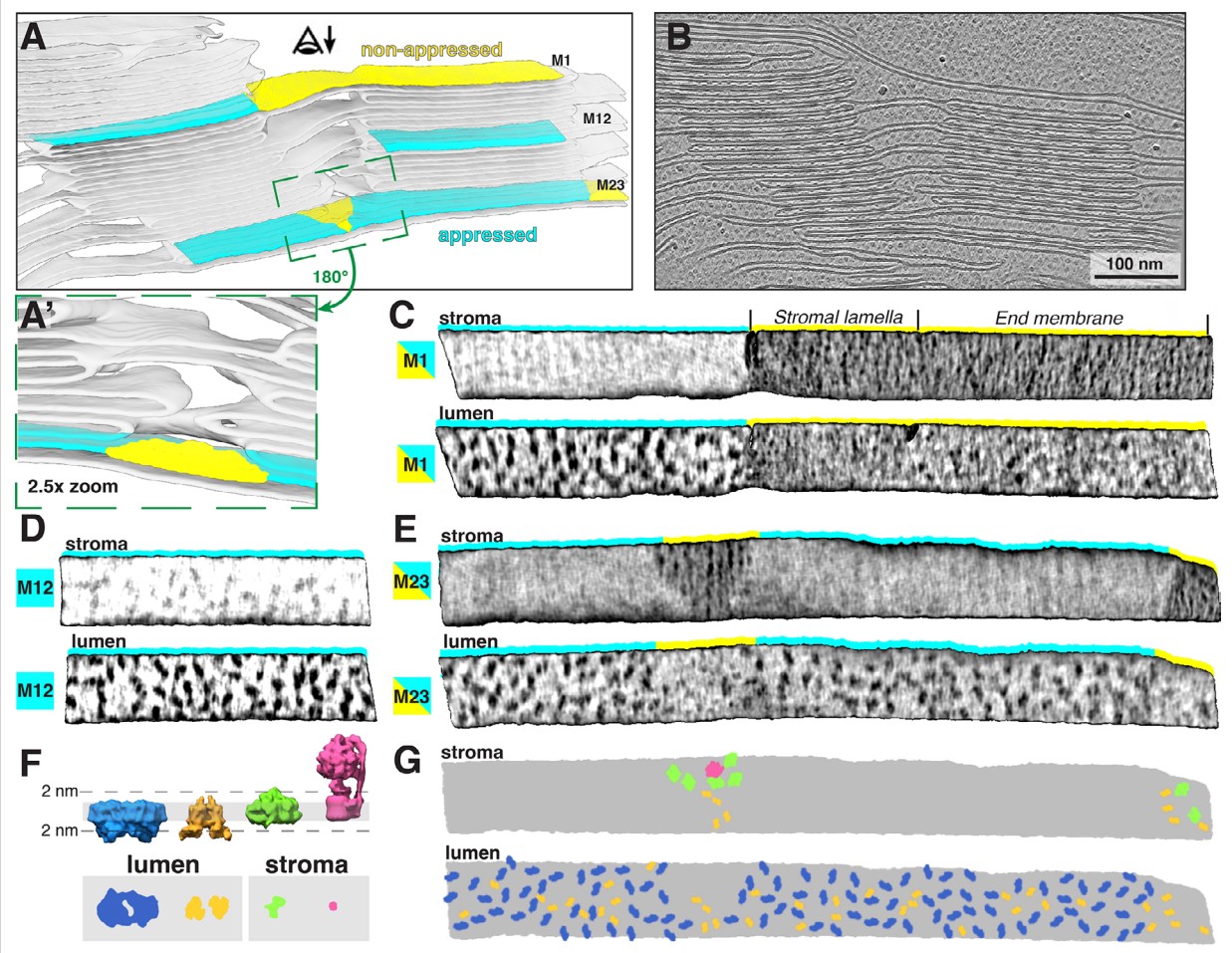

**Figure 5.** Lateral heterogeneity of PSII and PSI between appressed and non-appressed membranes. (**A**) Segmentation of the thylakoid network showing two close grana. Three membrane pieces are highlighted: M1, an appressed membrane of the granum (teal) transitions into a stromal lamella membrane (yellow) and then into an end membrane of the second granum (yellow); M12, an appressed membrane in the middle of the granum; M23, a membrane that spans two grana with a small unstacked stromal lamella in the middle (zoomed-in and rotated in green inset panel A') and at the right end. The eye with the arrow indicates the viewing direction in membranograms. (**B**) Slice through the corresponding tomographic volume. (**C, E**) Membranograms showing both the luminal and stromal sides of the three membranes highlighted in A. Color strips indicate the membrane domain (teal: appressed, yellow: non-appressed). Stromal lamellae and end membranes are annotated above the membranogram. All membranograms show densities 2 nm above the membrane surface (see F for reference). There are sharp boundaries between densities in the appressed and non-appressed regions. PSII densities are seen only on the luminal side of the appressed membranes, whereas PSI densities are seen only on the stromal side of non-appressed membranes. (**F**) Top: Diagram showing side views of structures (blue: PSII, orange: cyt$b_6$f, green: PSI, pink: ATPsyn) in a thylakoid membrane (gray). Bottom: membrane surface views showing densities that protrude from each structure 2 nm into the thylakoid lumen or stroma. (**G**) Cartoon representation of the stromal and luminal surfaces of the membrane in E, with particle footprints pasted in (colored as in F).

The online version of this article includes the following figure supplement(s) for figure 5:

**Figure supplement 1.** Protein densities on two domains of thylakoid membranes.

some particles (~2.1% overlap of total footprint area, *Figure 7A* second row, *7D*). The steric hindrance increased to ~8.2% overlap when larger $C_2S_2M_2$ supercomplexes were used (*Figure 7A* third row, *7D*). Depending on which footprint was mapped in, the total membrane surface coverage by the PSII cores and the PSII-LHCII supercomplexes ranged from ~20 to~55% (*Figure 7B*), leaving ample space in the membrane for additional free LHCII trimers and cyt$b_6$f. Our observation of some clash between $C_2S_2M_2$ supercomplexes suggests that different supercomplex variants probably coexist within the same membrane. Cumulative coverage of $C_2$, $C_2S_2$, and $C_2S_2M_2$ footprints when overlaying sequential membranes across the granum (known as the 'granum crossection') resulted in complete cross-section coverage after a minimum of six membranes (three consecutive thylakoids) in the case of the biggest supercomplex (*Figure 7C*).

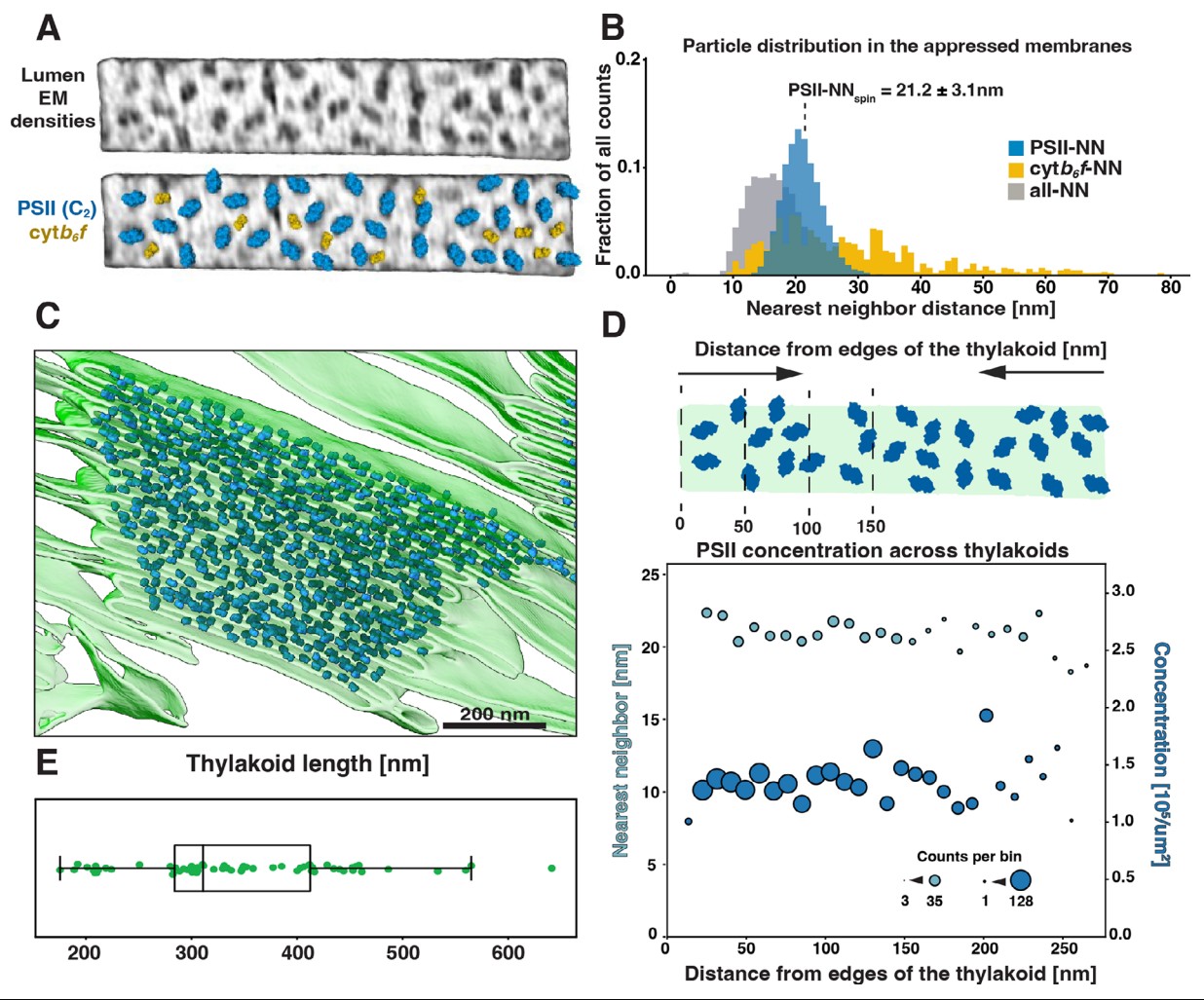

**Figure 6.** Distribution and concentration of PSII in grana membranes. (**A**) Top: membranogram view of an appressed membrane piece with characteristic densities corresponding to dimers of PSII and cyt$b_6f$. Bottom: same membrane with models of PSII and cyt$b_6f$ complexes superposed on the EM densities. (**B**) Nearest neighbor distance plot of all the particles from the appressed membranes. Blue: PSII, orange: cyt$b_6f$, gray: all particles. Note that 'all' also includes unassigned particles of unknown identity. The dotted line labeled PSII-NN$_{Spin}$ indicates the mean (± SEM) center-to-center distance between neighboring PSII cores. Nearest neighbor distances are much more variable between two cyt$b_6f$ than between two PSII complexes. Identification of cyt$b_6f$ particles was possible only in the best-resolved membranes; therefore, particle number and distribution estimation are not as reliable. (**C**) Segmentation of a granum with all detected PSII particles pasted in. Note that we did not analyze PSII distribution at the very tips of thylakoids; this is why the tips look unoccupied. (**D**) Plot showing PSII nearest neighbor center-to-center distances (light blue) and concentration (dark blue) as a function of the particle position from the edge of the granum, in 10 nm bins (diagrammed above the plot). Dot size represents number of counts in each bin. (**E**) Plot of the lengths of grana thylakoids selected for the analysis in D. Thylakoids of significantly different lengths were selected to minimize any potential effect of the membrane size on the PSII distribution.

We do not observe instances where PSII complexes directly face each other across the luminal gap (*Figure 8A and B*, *Figure 8—videos 1 and 2*). Rather, PSII densities interdigitate despite the thylakoid lumen being wide enough to accommodate two directly apposed complexes (*Figure 4*). Indeed, the luminal overlap is negligible between all visible densities (including cyt$b_6f$ and unknown particles) for the two membranes of an appressed grana thylakoid (e.g. M19-M20 and M21-M22 membrane pairs in *Figure 8B*). Similarly, across the stromal gap, the overlap of all densities is only slightly higher (~4%) (e.g. M20-M21 membrane pair in *Figure 8B*). Performing this stromal overlap quantification with supercomplex footprints (*Figure 7E*) showed that PSII cores have 9% of their surface area overlapping, whereas $C_2S_2$ and $C_2S_2M_2$ supercomplexes show 22% and 38% overlap of their area, respectively (*Figure 7F*). Within intact chloroplasts, we did not observe patterns of PSII supercomplexes aligning across appressed membranes (*Figure 8A and B*), contrary to the ordered lattices of PSII previously

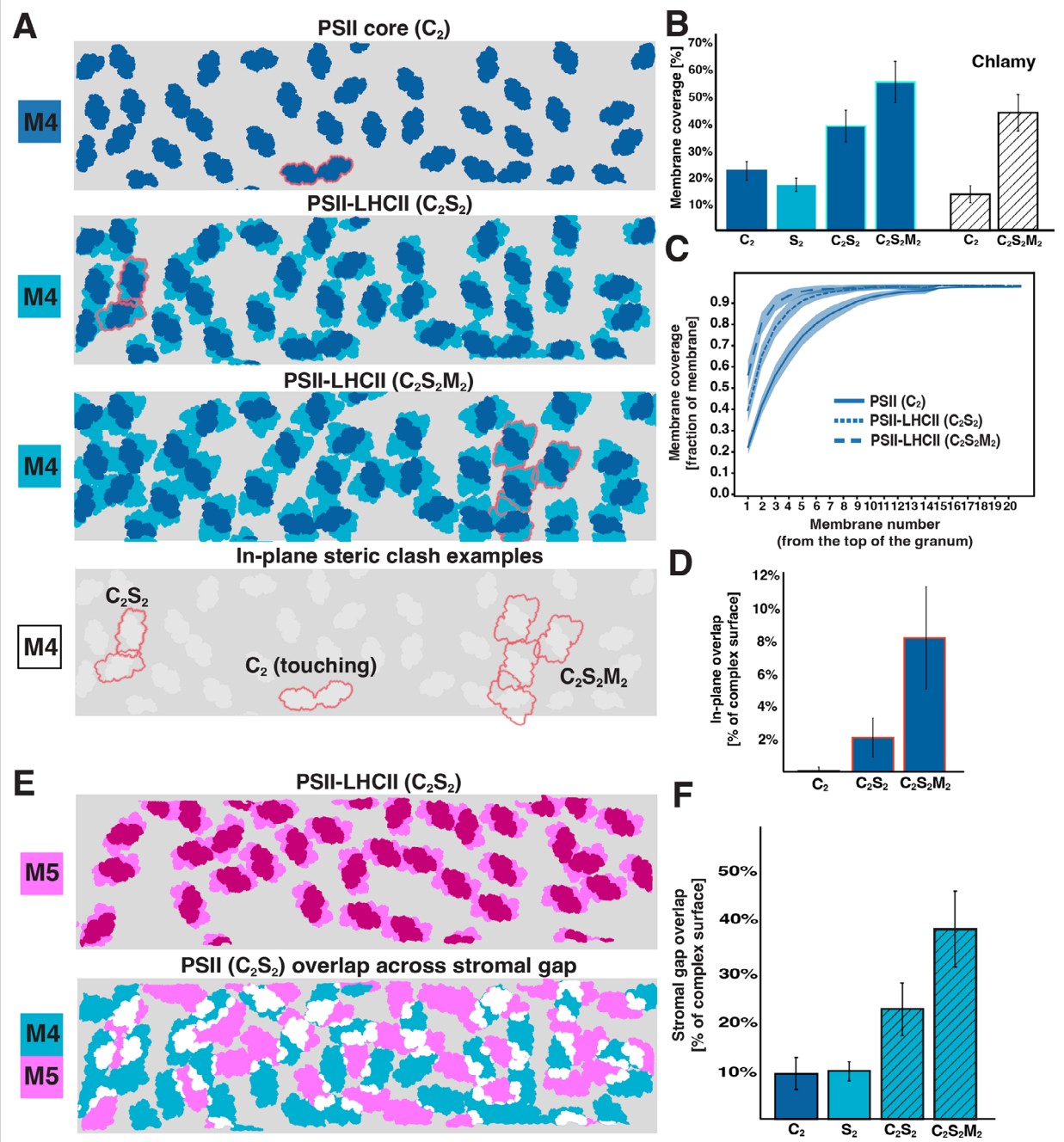

**Figure 7.** Occupancy of PSII and PSII-LHCII supercomplexes in grana membranes. (**A**) Top panel: cartoon representation of an analyzed membrane patch (M4, fourth membrane in a granum) with footprints of all PSII core complexes ($C_2$) placed into their respective positions. Middle panels: the same membrane patch with footprints of $C_2S_2$ and $C_2S_2M_2$ PSII-LHCII supercomplexes placed in. Bottom panel: red outlines highlight possible in-plane clashes between different types of supercomplexes within the same membrane (outlines carried down from panels above). (**B**) Quantification of the membrane coverage, depending on the type of supercomplexes placed in (the area of $S_2$ LHCII trimers in supercomplexes is also shown). Reproduced results from the *C. reinhardtii* study (Chlamy) shown for comparison (**Wietrzynski et al., 2020**). Error bars: SD. (**C**) Cumulative membrane coverage of PSII supercomplexes throughout the stack. Each line in the plot represents the percentage of area occupied by PSII or different PSII-LHCII supercomplexes when adding membranes across the stack (from the first stacked membrane until the end of the granum). Line: mean, shading: SEM. Membranes from 15 grana were used for the analysis. (**D**) Quantification of the in-plane steric clashes between PSII cores or different supercomplexes. PSII-PSII core clash: 0.0002% (within error range); these might be complexes in direct contact like the example in A. (**E**) Top: cartoon representation of the fifth membrane (M5) from the same granum with all $C_2S_2$ supercomplexes in place. Bottom, overlay of M4 (blue) and M5 (magenta) showing the overlap (white) between PSII-LHCII in appressed membranes separated by a stromal gap. (**F**) Quantification of the stromal overlap between PSII cores, LHCII $S_2$ trimers, and different supercomplex variants. For all plots, data was generated from 3 chloroplast preparations, using the best resolved membranes (n=173).

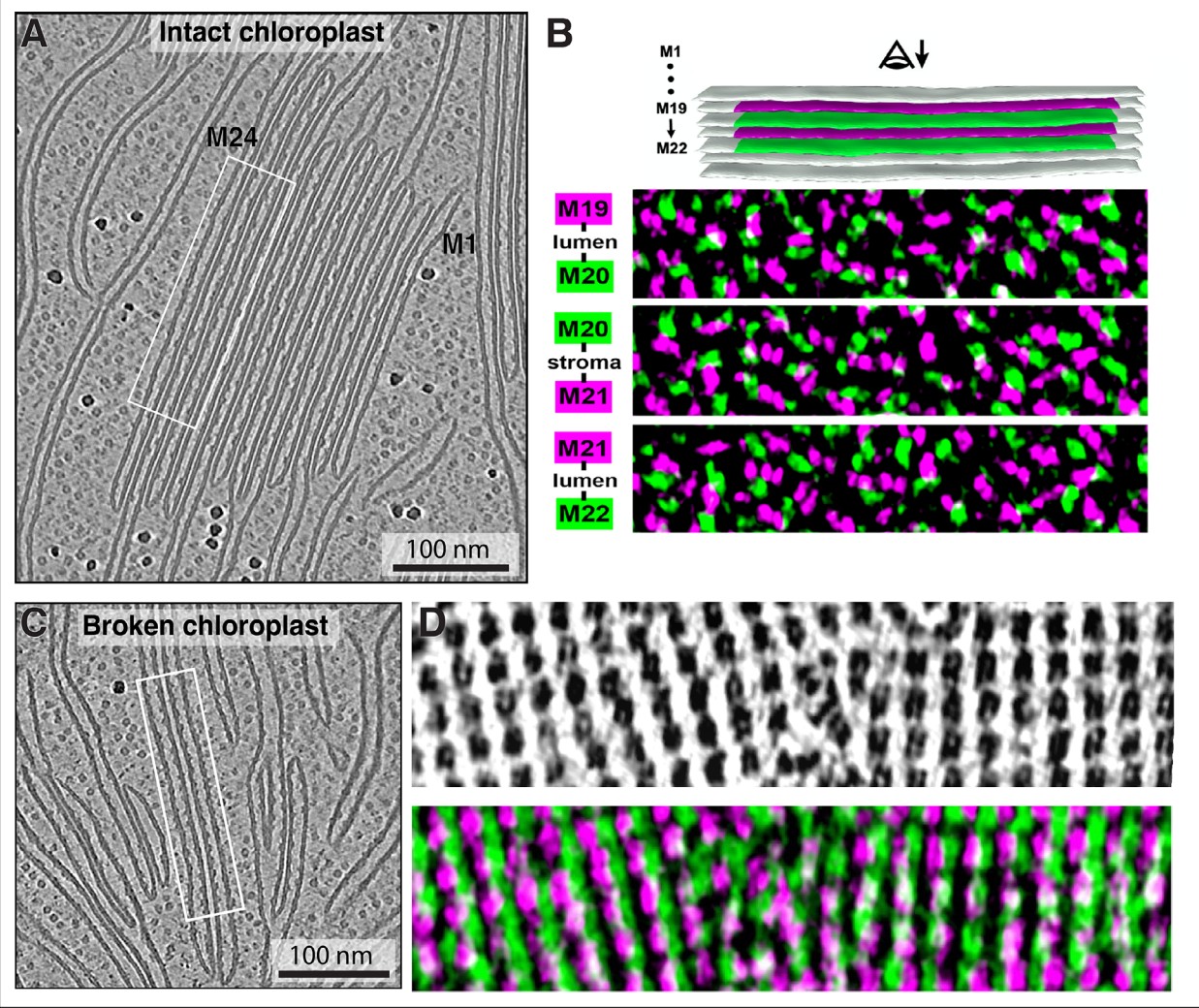

**Figure 8.** Overlap of densities across stacked membranes: intact vs. broken chloroplasts. (**A**) Tomographic slice showing a typical granum inside an intact chloroplast. White box marks the stack of membranes visualized in B. (**B**) Above: segmentation of the piece of a granum from A with four membranes highlighted (M19–M22). Odd-numbered membranes in magenta, even-numbered in green. Below: corresponding membranogram overlays of the adjacent membrane pairs. Coloring represents all EM densities protruding ~2 nm from the luminal side of each membrane. The three examples show two luminal overlaps (top and bottom) and an overlap across a stromal gap (middle). White: overlap between electron microscopy (EM) densities. (**C**) Tomographic slice showing a part of a broken chloroplast, containing individual and potentially re-stacked thylakoids. White box marks the adjacent thylakoids visualized in D. (**D**) Top: membranogram from a piece of one appressed membrane in C, showing densities protruding ~2 nm from the membrane's luminal surface. Note the arrays of PSII complexes and absence of any other type of particles. Bottom: membranograms from the pair of appressed membranes (green and magenta) overlaid with each other. White: overlap between EM densities. See *Figure 8—videos 1 and 2* for tomographic slices through the volumes shown in A and C.

The online version of this article includes the following video(s) for figure 8:

**Figure 8—video 1.** Slices through the tomogram shown in *Figure 8A*, highlighting the organization of PSII particles in near-native thylakoid membranes within an intact chloroplast.

https://elifesciences.org/articles/105496/figures#fig8video1

**Figure 8—video 2.** Slices through the tomogram shown in *Figure 8C*, highlighting the organization of PSII particles in thylakoid membranes from a broken chloroplast.

https://elifesciences.org/articles/105496/figures#fig8video2

observed in thylakoids isolated from pea (*Daum et al., 2010*). However, in tomograms of chloroplasts ruptured during blotting (e.g. *Figure 1—figure supplement 1F*), we did observe that PSII densities sometimes align into semi-crystalline lattices with a high degree of stromal overlap between the PSII complexes from adjacent appressed membranes (*Figure 8C and D*). It is, therefore, plausible

that extraction of thylakoids away from the chloroplast environment can induce the formation of PSII lattices between appressed membranes, some of which may have undergone restacking.

## Conclusions and speculation

Our cryo-ET study of intact spinach chloroplasts provides insights that span scales from the level of local membrane architecture (fine 3D ultrastructure of individual grana and connecting stromal lamellae) to the level of how individual protein complexes are organized within these membranes.

The architecture of spinach thylakoids visualized here is consistent with the previously proposed helical staircase models (*Mustárdy et al., 2008*; *Paolillo, 1970*; *Bussi et al., 2019*), although the structural details we observe are perhaps less stringent. The grana stacks of appressed membranes are not completely cylindrical: they have a wavy circumference (when viewed from the top, e.g. *Figure 3*), vary in diameter along the stack, and sometimes nearly merge with one another. We observe a variety of membrane architectures at the interconnections between appressed and non-appressed regions. Sometimes, the appressed domains shift smoothly into the non-appressed domains without architectural contortions (*Figure 5*, *Figure 5—figure supplement 1C*), similar to what was observed in *C. reinhardtii*. However, much more complex transitions were also observed, with narrow membrane bridges or sharp forks (*Figure 1—figure supplement 3*). At this level of membrane architecture, we also observed thylakoid contacts with plastoglobules (*Figure 2*) and the chloroplast envelope (*Figure 1—figure supplement 2*). The pristine sample preservation inside vitrified chloroplasts, combined with the high resolution of our cryo-ET imaging, allowed us to measure thylakoid morphometrics (bilayer thickness, stromal and luminal gaps) with sub-nanometer precision (*Figure 4*).

At the level of molecular organization, the appressed and non-appressed membranes have markedly different composition, clearly visualized by the protruding protein densities in membranograms (*Figure 5*, *Figure 5—figure supplement 1*). Similar to *C. reinhardtii* (*Wietrzynski et al., 2020*), there is a precise segregation between the two domains, with PSII restricted to the appressed membranes and PSI restricted to the non-appressed membranes. Cyt$b_6f$ is present in both domains. In contrast to previous biochemical studies, no obvious difference was seen between sub-regions of the non-appressed membranes (stromal lamellae, grana margins, grana end membranes) (*Figure 5C and E*; *Figure 5—figure supplement 1*). Similarly, our analysis of PSII distribution in the appressed grana membranes showed that these complexes are uniformly distributed through the stack (*Figures 6 and 7*), with no changes in concentration observed across the grana layers nor close to the granum edge, where it meets the non-appressed membrane region. This confirms that, at least on the level of PSII distribution, there are no organizational differences across the granum.

Together, these observations support a simple two-domain model for thylakoid lateral heterogeneity in vivo, which states that (1) the appressed and non-appressed domains each have a homogeneous mix of their respective proteins throughout, and (2) there is an abrupt and discrete transition between the domains, with no visible mixing between the PSI and PSII complexes. Nonetheless, the domain boundary (grana margin) must be sufficiently fluid to facilitate the exchange of protein complexes during dynamic processes such as PSII repair and state transitions.

In the intact chloroplasts, we did not observe ordered lattices of PSII that were previously reported in less intact samples from various species (*Daum et al., 2010*; *Levitan et al., 2019*; *Boekema et al., 2000*; *Simpson, 1983*). However, we did see semi-crystalline arrangements of PSII complexes in chloroplasts that had ruptured on the EM grid (*Figure 8*, *Figure 8—videos 1 and 2*). Rearrangement of PSII into ordered arrays may, therefore, be triggered by exposure of the thylakoid network to a non-native environment with different buffering properties (*Staehelin, 1976*; *Tsvetkova et al., 1995*; *Ojakian and Satir, 1974*). It remains to be determined whether PSII lattices could also have a physiological function in vivo (*Eckardt et al., 2024*) (to the best of our knowledge, the only in situ report of PSII arrays is in the dehydrated state of the resurrection plant *Craterostigma pumilum*; see *Charuvi et al., 2016*).

In situ cryo-ET only provides snapshots of living systems, but perhaps we can infer hints about the dynamic nature of thylakoids from these frozen moments in time. Our results show that PSII-LHCII supercomplexes do not align with each other under standard growth conditions. This distribution is consistent with supercomplexes mixing freely within the appressed domains without making persistent static interactions to other supercomplexes. Based on evidence that LHCII plays an important role in thylakoid stacking (*Daum et al., 2010*; *Armond et al., 1977*; *McDonnel and*

*Staehelin, 1980*; *Albanese et al., 2020*; *Guardini et al., 2022*; *Standfuss et al., 2005*), we hypothesize that while LHCII trimers can make relatively stable in-plane interactions to form PSII-LHCII supercomplexes, the other interactions within an appressed membrane and across the stromal gap to an adjacent stacked membrane are more transient. LHCII trimers are intrinsically multivalent and make electrostatic interactions with each other across the stromal gap using their disordered, positively-charged N-terminus (*Standfuss et al., 2005*). Such a network of weak and multivalent interactions is reminiscent of the organizing principles of liquid-liquid phase separation (*Mittag and Pappu, 2022*; *Snead and Gladfelter, 2019*). We speculate that stacked thylakoids in both green algae and plants are a condensation of membrane-embedded PSII-LHCII supercomplexes and free LHCII proteins. This condensation is driven by an ensemble of interactions both within the membrane plane and between stacked membranes (the latter via the N-terminus of LHCII), which are modulated by macromolecular crowding effects and membrane surface charge, respectively (*Ojakian and Satir, 1974*; *Izawa and Good, 1966*; *Kim et al., 2005*). In contrast to canonical globular condensates, we propose that thylakoid stacks are multiple interacting layers of planar condensates. In addition to being a main driving force for the condensation of LHCII and PSII, these interactions between membrane layers also likely provide the mechanism for sterically excluding ATPsyn and PSI from the condensate, while permitting entry of cyt$b_6f$ because it lacks a large stromal density. An important prediction of this stacking condensation model is that PSII, LHCII, and cyt$b_6f$ are mobile within the appressed regions; however, we lack the methodology to directly track movements of single protein complexes within a native grana membrane.

## Limitations and future perspectives

Plant cells are generally difficult to vitrify and are too large for conventional FIB milling workflows. We opted for chloroplast extraction and plunge-freezing (*Figure 1—figure supplement 1*), which facilitated the preparation, however, not without disadvantages. The minutes-long isolation protocol can lead to relaxation of the physiological state of the chloroplasts, especially if it is temperature- or light-dependent. Moreover, plunge-freezing on EM grids can rupture the chloroplast envelope (*Figure 1—figure supplement 1B, F*) and promote preferential orientation of the plastids — most often the entire thylakoid network lies flat on the surface of the grid support. Flat membranes, orthogonal to the electron beam during cryo-ET acquisition, provide top views (*Figure 3*, *Figure 3—figure supplement 1*) that suffer from low signal and are difficult to segment and analyze. We collected most of the tomograms towards the edges of the chloroplasts where this effect is less pronounced (more frequent side views), but this approach could induce region-specific bias.

Although more technically demanding, recent advances in cryo-FIB lift-out of high-pressure-frozen multicellular samples (*Schiøtz et al., 2024*; *Nguyen et al., 2024*) open future possibilities to perform cryo-ET on chloroplasts within native leaf tissue. Still, the relatively low throughput of cryo-ET and intrinsically small volume imaged by this technique present challenges for answering micron-scale questions about the overall membrane ultrastructure of the chloroplast; those questions are being successfully tackled by other microscopy techniques that image larger sample volumes at the trade-off of resolution (e.g. *Shimoni et al., 2005*; *Bos et al., 2023*; *Mustárdy et al., 2008*; *Bussi et al., 2019*; *Paolillo, 1970*; *Iwai et al., 2018*; *Pipitone et al., 2021*). A more complete picture of thylakoid networks across length and time scales will require integrating the native molecular views provided by cryo-ET with complementary structural and cellular techniques (*McCafferty et al., 2024*). Thylakoids are dynamic systems that adapt their molecular architecture to modulate their function in response to environmental changes. A future challenge is to accurately replicate this complexity with mesoscale modeling that incorporates parameters from diverse experiments to bridge thylakoid structure and function.

# Materials and methods

**Key resources table**

| Reagent type (species) or resource | Designation | Source or reference | Identifiers | Additional information |
|---|---|---|---|---|
| Biological sample (*Spinacia oleracea*) | Common Spinach plants, WT strain | LMU Munich Plant facility stock | | Freshly isolated chloroplast from the 5–6- weeks-old plants |
| Other | Autogrids | Thermo Fisher Scientific | 1205101 | Cryo-EM material |
| Other | Teflon Sheets for blotting | plastx24.de | 11645 | Cryo-EM material |
| Other | Whatman filter paper for blotting | Sigma Aldrich | 10311807 | Cryo-EM material |
| Other | C-rings | Thermo Fisher Scientific | 1036171 | Cryo-EM material |
| Other | EM Grids R2/1 carbon | Quantifoil | N1-C15nCu20-01 | Cryo-EM material |
| Software, algorithm | MemBrain v1 | *Lamm et al., 2022*; *Lamm, 2023* | | https://github.com/CellArchLab/MemBrain |
| Software, algorithm | MemBrain v2 | *Lamm et al., 2024*; *Lamm, 2025* | | https://github.com/CellArchLab/MemBrain-v2 |
| Software, algorithm | TOMOMAN | *Khavnekar et al., 2024*; *Wan, 2025* | | https://github.com/wan-lab-vanderbilt/TOMOMAN |
| Software, algorithm | The Virtual Brain | *Sanz Leon et al., 2013* | | https://github.com/the-virtual-brain |
| Software, algorithm | PyVista | *Sullivan and Kaszynski, 2019* | | https://doi.org/10.21105/joss.01450 |
| Software, algorithm | Membranorama | *Tegunov, 2023* | | https://github.com/dtegunov/membranorama |
| Software, algorithm | DeepDeWedge | *Wiedemann and Heckel, 2024*; *Wiedemann, 2025* | | https://github.com/MLI-lab/DeepDeWedge |
| Software, algorithm | Cryo-CARE | *Buchholz et al., 2019*; *Wagner, 2024* | | https://github.com/juglab/cryoCARE_pip |
| Software, algorithm | IMOD v4.11.1 | *Mastronarde and Held, 2017*; *LabShare-Archive, 2018* | | https://github.com/LabShare-Archive/IMOD |
| Software, algorithm | MotionCor2 v1.4.0 | *Zheng et al., 2017*; *Han, 2017* | | https://github.com/singleparticle/MotionCor2 |
| Software, algorithm | TomoSegMemTV | *Martinez-Sanchez et al., 2014* | | https://sites.google.com/site/3demimageprocessing/tomosegmemtv |

## Plant growth

*Spinacia oleracea* WT plants were germinated and grown in a growth chamber for 6 weeks with a 12 hr day/night cycle at 21–23°C and white-light illumination (150 µmol photons $m^{-2}s^{-1}$). One day before the experiment, the night cycle was extended to 20 hr to minimize starch content in the chloroplasts. Prior to chloroplast isolation, plants were transferred from darkness to illumination for 1 hr to acclimate them to light. This experiment was performed in three independent batches.

## Chloroplast isolation

Approximately 80 grams of fresh leaf mass was collected and immediately blended with 160 mL of isolation buffer (0.45 M Sorbitol, 20 mM Tricine-KOH, 10 mM EDTA, 10 mM $NaHCO_3$, 5 mM $MgCl_2$, 0.1% BSA, 0.2% D-Ascorbate) on ice. The slurry was filtered twice through cotton and a double layer of Miracloth to remove debris. The filtrate was then deposited on a 50% Percoll cushion in isolation buffer and centrifuged for 8–10 min at 7500 rpm. The green band in between percoll layers was collected and diluted in the isolation buffer at least three times (dilution depended on the efficiency of purification and type of the EM grids to be used). Chloroplasts were plunge-frozen immediately afterwards.

## Plunge-freezing

Plunging was performed using a Vitrobot Mark 4 (Thermo Fisher Scientific). 4 µL of chloroplasts solution was placed onto R2/1 or R1.2/1.3 carbon-coated 200-mesh copper EM grids (Quantifoil Micro

Tools), blotted for 7–8 s from the back side (front side pad covered with a Teflon sheet) with blot-force 10, and plunge-frozen in a liquid ethane/propane mixture cooled with liquid nitrogen. Grids were clipped into 'autogrid' support rings modified with a cut-out on one side (FEI, Thermo Fisher Scientific) and stored in plastic boxes in liquid nitrogen until used for cryo-FIB milling.

## Focused ion beam milling

Cryo-FIB milling was performed as described previously (*Schaffer et al., 2017*) using an Aquilos dual-beam FIB/SEM instrument (Thermo Fisher Scientific). Grids were screened for good chloroplast coverage and coated with a layer of organometallic platinum using a gas injection system to protect the sample surface. The sample was milled with a gallium ion beam to produce ~100–150 nm-thick lamellae. Some batches of lamellae were additionally sputter-coated with platinum at the end of the milling process. Grids were then transferred in liquid nitrogen to the TEM microscope for tomogram acquisition.

## Cryo-ET data acquisition

Tomographic data was collected on a 300 kV Titan Krios TEM instrument (Thermo Fisher Scientific), equipped with a post-column energy filter (Quantum, Gatan), and a direct detector camera (K2 Summit, Gatan). Tilt series were acquired using SerialEM software (*Mastronarde, 2005*) with a dose symmetric scheme (*Hagen et al., 2017*) starting at the pretilt of 10/–10° (depending on the loading direction of the grids), with 2° increment between tilts and spanning a total of 120°. Individual tilts were recorded in counting mode with an imaging rate of 8–12 frames per second, at a pixel size of 3.52 Å and target defocus in the range of 2.5–5 μm. The total accumulated dose for the tilt series was kept at approximately 120 e-/Å$^2$.

## Tomogram reconstruction and processing

All tilt-series were processed using the TOMOMAN pipeline (*Khavnekar et al., 2024*). Raw frames were aligned using MotionCor2 (v.1.4.0) (*Zheng et al., 2017*). Bad tilts were manually removed, and cleaned tilt series were dose-weighted. The assembled tilt-series were aligned with patch tracking using the Etomo program from the IMOD (v.4.11.1) package (*Kremer et al., 1996*; *Mastronarde and Held, 2017*). Final tomograms were reconstructed by weighted back projection in IMOD. To enhance contrast, the cryo-CARE denoising was applied (*Buchholz et al., 2019*). For the views in *Figure 3*, we instead performed denoising with DeepDeWedge, which additionally attempts to fill in missing wedge information (*Wiedemann and Heckel, 2024*). All images of tomograms (tomographic slices) shown were acquired using the IMOD 3Dmod viewer.

## Membrane segmentation

Membrane segmentation was performed using AI-assisted approaches as described in *Lamm et al., 2024*. Briefly, in order to generate membrane segmentations, we initially segmented the tomograms using the TomoSegMemTV segmentation protocol (*Martinez-Sanchez et al., 2014*), and extracted patches of 160$^3$ voxels of tomogram-segmentation pairs. After manually cleaning these segmented patches, we used these pairs to train a neural network (nnU-Net) to segment the entire thylakoid lattice. We predicted segmentations using the trained network, extracted and corrected new membrane patches, merged these into our training set, and started a new round of training. We followed this iterative approach until we were satisfied with the performance of our segmentation model. This model is now available in the MemBrain-seg package (*Lamm et al., 2024*). All segmentations shown in the figures were generated using this model, then cleaned in Amira (Thermo Fisher Scientific) and rendered in UCSF ChimeraX.

## Preparation of single-membrane segmentations

For all analyses requiring single membrane instances or stacks of individual membranes (e.g. particle organization in the membrane and overlays between membranes), the separate segmentations were made by cutting the full-tomogram segmentations from MemBrain-seg in Amira and exporting them as separate volume files (.mrc) and triangulated surface meshes.

## Measurement of appressed vs. non-appressed areas

For the quantification of appressed vs. non-appressed regions (shown in *Figure 1E and F*), we first segmented all membrane networks in all our tomograms using our trained MemBrain-seg model and then manually cleaned the output segmentations in Amira to remove tomogram edge artifacts. Afterwards, we converted the segmentations into triangular meshes using PyVista's marching cubes algorithm (*Sullivan and Kaszynski, 2019*), followed by Laplacian smoothing, giving normal vectors for each triangle on the mesh. For each triangle, we traced the distance to the next triangle along the normal vector. This next triangle along the normal belongs to potentially neighboring membranes, and by thresholding the distance, we divided the meshes into 'appressed' and 'non-appressed' triangles. To account for noisy distance predictions, we applied a majority vote among geodesically neighboring triangles to receive a smooth representation of appressed and non-appressed regions. Furthermore, by computing the principal curvature at each triangle, we excluded the membrane segmentation edges (e.g. clipped by the tomogram volume or ending due to mis-segmentation), as these have normal vectors pointing away from the stack and, therefore, would distort the metrics towards more non-appressed regions. By summing up the triangle areas of all remaining mesh triangles for both classes, we received the total areas of appressed and non-appressed membranes.

## Thylakoid spacing and thickness calculations

In order to measure distances between different membrane layers, stromal and luminal gaps, as well as entire thylakoid thickness (shown in *Figure 4*), we utilized membrane flattening functionalities from blik (*Gaifas et al., 2024*). The goal of this step was to receive a 'stack' of 2D images for each membrane, where each 2D image in the z-stack corresponds to the flattened front view of the membrane at a different distance along the membrane normal vector. This is to remove the effects of membrane curvature and increase signal-to-noise ratio of the measurements, as well as to ensure sampling orthogonal to the plane of lipid bilayers. To perform the flattening, we first sampled a grid of points on each 3D membrane segmentation. This grid allowed blik to generate the 2D image stacks along the normal vectors. Next, we divided each membrane into 50 nm patches. By averaging the intensities of each 2D image in these stacks, we received a single value for each z-coordinate, leading to a 1D plot along the z-axis, i.e., the membrane normal vector. From this 1D plot, we extracted local intensity minima and maxima corresponding to center points of lipid bilayers. After manual cleaning of outlier plots (i.e. plots where the automated assignment of keypoints failed), this representation allowed us to estimate the distances between different membranes and gaps. To account for the soft edge of the membrane in cryo-ET data, we selected a halfway point between the maximal signal of the membrane leaflet to the center of the membrane as the 'edge' and applied this correction symmetrically on both sides of the signal peak for more accurate measurements. See *Figure 4— figure supplement 1* for the visual description.

## Membrane protein ground truth positions

For training MemBrain, our membrane protein localization network (*Lamm et al., 2024*), we required ground truth positions for the PSII proteins. To generate them, we imported the single membrane. obj files into Membranorama (https://github.com/dtegunov/membranorama, *Tegunov, 2023*) to visualize the membranes and their embedded proteins by projecting the tomogram densities onto the surface mesh. In this program, we were able to manually annotate PSII positions and orientations in 45 membrane patches from seven different tomograms, which we then used as training data for the MemBrain model.

## MemBrain training

We utilized our membrane protein ground truth data to train a MemBrain model. To do this, we split our full dataset consisting of 45 membranes into a training set (28 membranes), a validation set (7 membranes), and a test set (10 membranes). As a required first step in the MemBrain workflow, we manually defined the side of each membrane facing toward the thylakoid lumen as the side of interest. Then, we trained MemBrain using its default parameters to predict 'heatmaps' depicting each membrane point's distance to the closest PSII. Using our ground truth annotations (validation set), we further tuned the best bandwidth parameter for the subsequent Mean Shift clustering, which we determined to be 21 bin4 voxels, corresponding to 29.6 nm.

## Application of trained MemBrain model

After training and tuning MemBrain, we applied it to the full dataset, consisting of 455 discrete membrane segmentations from the nine best quality tomograms. This gave us good initial predictions for positions and orientations of PSII complexes in all membranes. In order to confirm their validity, we inspected all 455 membranes again, discarded 152 membranes due to insufficient quality, and manually corrected false MemBrain predictions. The remaining 303 membranes were used for the analysis. Wherever image quality allowed, we also annotated $cytb_6f$ positions. This gave us an accurate membrane protein dataset, consisting of 14433 PSII, 937 $cytb_6f$, and 5934 positions of unknown particles (non-identifiable densities).

## Center-to-center distances

In order to analyze the nearest neighbor distances between our particle positions, we first converted each voxel-wise membrane segmentation into a surface mesh representation using PyVista's marching cubes algorithm, followed by Laplacian surface smoothing (applied to individual single membrane instances). This allowed us to compute exact geodesic distances between two positions on the mesh using The Virtual Brain's (https://github.com/the-virtual-brain, *Sanz Leon et al., 2013*) implementation of the discrete geodesic distance algorithm (*Mitchell et al., 1987*). For each particle location, we computed the geodesic distance to the center position of the nearest neighbor from the respective target particle class (e.g. PSII-NN$_{Spin}$, $cytb_6f$-NN$_{Spin}$, all-NN$_{Spin}$; shown in *Figure 6B*).

## Distance to appressed region edge

To analyze the change of protein distribution with respect to distance from the edge of appressed membrane regions, we first annotated these membrane edges manually using Membranorama. We then divided the Euclidean distance to the membrane edge into 10 nm-sized bins and computed the total area of mesh triangles within each bin. By dividing the number of particle positions by the triangle area within each bin, we obtain average protein complex concentrations per bin. Analogously, for each bin, we computed the average nearest neighbor distances. Plotting these concentrations against the respective bin distances provided the graphs displayed in *Figure 6D*.

## Overlap analysis

To analyze the overlap between PSII-LHCII supercomplexes in neighboring membranes (*Figure 7E and F*) or cumulative occupancy in membranes across the stack (*Figure 7C*), we projected protein positions and orientations onto a common membrane surface. We calculated their overlaps by comparing the overlaying triangles on the segmented membrane. To process a stack of membranes, we calculated a connection vector between the first and last membrane of the stack by computing the average membrane normal vector for the starting membrane. Then, for each membrane, we projected the corresponding membrane protein positions along the normal vector onto the starting membrane mesh. Mapping the protein complex footprint (based on its 3D structure filtered to 14 Å resolution) into the correct orientation in this new position allowed us to calculate the area on the membrane surface that was occupied by the respective complexes. Doing this for all membranes gave us a set of occupied areas per membrane, which we then analyzed for overlapping areas. To avoid bias due to membrane misalignment (e.g. stack being inclined in the tomogram slab resulting in incomplete overlap between membranes), we also projected the segmentations of all membranes onto the starting mesh and only analyzed the areas that were shared by all projected membrane segmentations.

## Acknowledgements

We thank the Plant and Greenhouse facility of LMU Munich and especially Anja Schneider for help with the cultivation of plants. We thank Jürgen Plitzko and Wolfgang Baumeister at the Max Planck Institute of Biochemistry for access to electron microscopes. We thank Sebastian Ziegler and Fabian Isensee from the Applied Computer Vision Lab (ACVL) at the German Cancer Research Center (DKFZ) for their support in designing the membrane segmentation framework. L L also thanks Julia Schnabel from the Institute of Machine Learning in Biomedical Imaging (IML) at Helmholtz Munich for advice and supervision. Cryo-ET analysis was performed at the sciCORE (http://scicore.unibas.ch/) scientific

computing center at the University of Basel. B D E and W W acknowledge funding from a Human Frontier Science Program (HFSP) research grant (award number RGP0005/2021) and European Research Council (ERC) consolidator grant 'cryOcean' (fulfilled by the Swiss State Secretariat for Education, Research and Innovation, award number M822.00045). M P J was supported by the grants from the Biotechnology and Biological Sciences Research Council (BBSRC) (award number BB/V006630/1) and the Leverhulme Trust (award numbers RPG-2019–045 and RPG-2021–345). L M was supported by a BBSRC White Rose DTP studentship in Mechanistic Biology. L L acknowledges support from the Munich School for Data Science (MUDS) and a fellowship from the Boehringer Ingelheim Fonds.

## Additional information

### Funding

| Funder | Grant reference number | Author |
|---|---|---|
| Biotechnology and Biological Sciences Research Council | BB/V006630/1 | Matthew P Johnson |
| Leverhulme Trust | RPG-2019-045 | Matthew P Johnson |
| State Secretariat for Education, Research and Innovation | M822.00045 | Benjamin D Engel Wojciech Wietrzynski |
| Human Frontier Science Program | 10.52044/hfsp. rgp00052021.pc.gr.165228 | Benjamin D Engel Wojciech Wietrzynski |
| Leverhulme Trust | RPG-2021-345 | Matthew P Johnson |
| Boehringer Ingelheim Fonds | | Lorenz Lamm |
| Munich School for Data Science | | Lorenz Lamm |
| Biotechnology and Biological Sciences Research Council | White Rose DTP | Lorna Malone |

The funders had no role in study design, data collection and interpretation, or the decision to submit the work for publication.

### Author contributions

Wojciech Wietrzynski, Conceptualization, Formal analysis, Investigation, Visualization, Writing – original draft, Writing – review and editing; Lorenz Lamm, Conceptualization, Data curation, Formal analysis, Validation, Investigation, Writing – original draft, Writing – review and editing; William HJ Wood, Matina-Jasemi Loukeri, Lorna Malone, Investigation; Tingying Peng, Supervision; Matthew P Johnson, Conceptualization, Supervision, Funding acquisition, Writing – original draft, Writing – review and editing; Benjamin D Engel, Conceptualization, Supervision, Funding acquisition, Writing – original draft, Project administration, Writing – review and editing

### Author ORCIDs

Wojciech Wietrzynski ![ORCID] https://orcid.org/0000-0001-8898-2392
Lorenz Lamm ![ORCID] https://orcid.org/0000-0003-0698-7769
William HJ Wood ![ORCID] https://orcid.org/0000-0003-0683-8085
Matina-Jasemi Loukeri ![ORCID] https://orcid.org/0009-0006-9172-5649
Tingying Peng ![ORCID] https://orcid.org/0000-0002-7881-1749
Matthew P Johnson ![ORCID] https://orcid.org/0000-0002-1663-0205
Benjamin D Engel ![ORCID] https://orcid.org/0000-0002-0941-4387

Reviewer #1 (Public review): https://doi.org/10.7554/eLife.105496.3.sa1
Reviewer #2 (Public review): https://doi.org/10.7554/eLife.105496.3.sa2

Author response https://doi.org/10.7554/eLife.105496.3.sa3

# Additional files

## Supplementary files
MDAR checklist

## Data availability
The raw cryo-ET data, cryo-CARE denoised tomograms and selected segmentation volumes are available at the Electron Microscopy Public Image Archive (EMPIAR), accession code: EMPIAR-12612. Positions of PSII particles used in the study are available in .star format at: 10.5281/zenodo.15090119. Denoised tomograms and segmentations shown in the figures are deposited in the Electron Microscopy Data Bank (EMDB), accession codes: EMD-52542, EMD-52543, EMD-52544, EMD-52545, EMD-52546, EMD-52547, EMD-52548.

The following datasets were generated:

| Author(s) | Year | Dataset title | Dataset URL | Database and Identifier |
|---|---|---|---|---|
| Wietrzynski W, Lamm L, Diogo Righetto R, Engel B | 2025 | Cryo-ET Spinach Thylakoid Photosystem II Positions | https://doi.org/10.5281/zenodo.15090119 | Zenodo, 10.5281/zenodo.15090119 |
| Wietrzynski W, Johnson MP, Engel BD | 2025 | Molecular architecture of thylakoid membranes within intact spinach chloroplasts | https://www.ebi.ac.uk/empiar/EMPIAR-12612/ | EMPIAR, EMPIAR-12612 |
| Wietrzynski W, Lamm L, Engel BD | 2025 | Cryo-electron tomogram of intact spinach chloroplast at 14.08A bin4 pixel size; denoised with cryoCARE to improve contrast | https://emdataresource.org/EMD-52542 | Electron Microscopy Data Bank, EMD-52542 |
| Wietrzynski W, Lamm L, Engel BD | 2025 | Cryo-ET tomogram #5 of thylakoids inside spinach chloroplast (top view on the membranes) | https://emdataresource.org/EMD-52543 | Electron Microscopy Data Bank, EMD-52543 |
| Wietrzynski W, Lamm L, Engel BD | 2025 | Cryo-electron tomogram #32 of thylakoids in spinach chloroplast. Denoised with CryoCARE | https://emdataresource.org/EMD-52544 | Electron Microscopy Data Bank, EMD-52544 |
| Wietrzynski W, Lamm L, Engel BD | 2025 | Cryo-electron tomogram #12 of thylakoids in spinach chloroplast. Denoised with cryoCARE | https://emdataresource.org/EMD-52545 | Electron Microscopy Data Bank, EMD-52545 |
| Wietrzynski W, Lamm L, Engel BD | 2025 | Cryo-electron tomogram of thylakoids in spinach chloroplast | https://emdataresource.org/EMD-52546 | Electron Microscopy Data Bank, EMD-52546 |
| Wietrzynski W, Lamm L, Engel BD | 2025 | Cryo-electron tomogram of thylakoids from broken spinach chloroplast. Denoised with cryoCARE | https://emdataresource.org/EMD-52547 | Electron Microscopy Data Bank, EMD-52547 |
| Wietrzynski W, Lamm L, Engel BD | 2025 | Cryo-electron tomogram #01 of thylakoids in spinach chloroplast. Denoised with cryoCARE | https://emdataresource.org/EMD-52548 | Electron Microscopy Data Bank, EMD-52548 |

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
