## [Editor Report · eLife Assessment]

The macromolecular organization of photosynthetic complexes within the thylakoids of higher plant chloroplasts has been a topic of significant debate. Using in situ cryo-electron tomography, this study reveals the native thylakoid architecture of spinach thylakoid membranes with single-molecule precision. The experimental methods are unique and **compelling**, providing **important** information for understanding the structural features that impact photosynthetic regulation in vascular plants and addressing several long-standing questions about the organization and regulation of photosynthesis.

---

## [Referee Report · Reviewer #1 (Public review)]

Summary:

In this study, the authors utilized in situ cryo-electron tomography (cryo-ET) to uncover the native thylakoid architecture of spinach chloroplasts and mapped the molecular organization of these thylakoids with single-molecule resolution. The obtained images show the detailed ultrastructural features of grana membranes and highlight interactions between thylakoids and plastoglobules. Interestingly, despite the distinct three-dimensional architecture of vascular plant thylakoids, their molecular organization closely resembles that of green algae. The pronounced lateral segregation of PSII and PSI was observed at the interface between appressed and non-appressed thylakoid regions, without evidence of a specialized grana margin zone where these complexes might intermix. Furthermore, unlike isolated thylakoid membranes, photosystem II (PSII) did not form a semi-crystalline array and was distributed uniformly within the membrane plane and across stacked grana membranes in intact chloroplasts. Based on the above observations, the authors propose a simplified two-domain model for the molecular organization of thylakoid membranes, which can be applied to both green algae and vascular plants. This study suggests that the general understanding of the functional separation of thylakoid membranes in vascular plants requires reconsideration.

Strengths:

By employing and refining AI-driven computational tools for the automated segmentation of membranes and identification of membrane proteins, this study successfully quantifies the spatial organization of photosynthetic complexes both within individual thylakoid membranes and across neighboring stacked membranes.

Weaknesses:

This study's weakness is that it requires the use of chloroplasts isolated from leaves and the need to freeze them on a grid for observation. However, the authors have correctly identified the limitations of this approach and have made some innovations, such as rapid sample preparation. The reliability of the interpretation of the results in light of previous results can be evaluated as high.

Comments on revised version:

The author has responded appropriately to the peer review comments and revised the paper.

---

## [Referee Report · Reviewer #2 (Public review)]

Summary:

For decades, the macromolecular organization of photosynthetic complexes within the thylakoids of higher plant chloroplasts has been a topic of significant debate. Using focused ion beam milling, cryo-electron tomography, and advanced AI-based image analysis, the authors compellingly demonstrate that the macromolecular organization in spinach thylakoids closely mirrors the patterns observed in their earlier research on *Chlamydomonas reinhardtii*. Their findings provide strong evidence challenging long-standing assumptions about the existence of a 'grana margin'-a region at the interface between grana and stroma lamellae domains that was thought to contain intermixed particles from both areas. Instead, the study establishes that this mixed zone is absent and reveals a distinct, well-defined boundary between the grana and stroma lamellae.

Strengths:

By situating high-resolution structural data within the broader cellular context, this work contributes valuable insights into the molecular mechanisms governing the spatial organization of photosynthetic complexes within thylakoid membranes.

Comments on revised version:

All reviewer comments have been fully addressed, and I have no further comments.

---

## [Author Response]

The following is the authors’ response to the original reviews.

**Reviewer #1 (Public review):**
Weaknesses:This study's weakness is that it requires the use of chloroplasts isolated from leaves and the need to freeze them on a grid for observation, so it is unclear to what extent the observations reflect physiological conditions. In particular, the mode of existence of the thylakoid membrane complexes seems to be strongly influenced by the physicochemical environment surrounding the membranes, as indicated by the different distribution of PSII between intact chloroplasts and those with ruptured envelope membranes.

We agree with the reviewer, as discussed in the “Limitations and Future Perspectives” section of our manuscript. The duration and conditions of the chloroplast isolation will very likely influence the state of the sample and hamper conclusions about physiological adaptations to environmental conditions, which are important for a dynamic process like photosynthesis. Isolated chloroplasts were the most feasible option for vitrification by plunge freezing, but we intend to improve our technological approaches to overcome this obstacle in the future (e.g., by using the more involved approach of cryo-lift out from high-pressure frozen tissue). Here, we hope that by using plants acclimated to a “standard state” (standard growth conditions under low light) and proceeding with fast isolation and grid preparation (chloroplast were used only once per isolation and deposited on the grids as fast as 10 min from leaf harvesting), we preserve some physiological relevance. This is supported by: (1) a PSII distribution pattern and concentration that is similar to previous observations by us and others in cryo-ET of FIB-milled algae cells and freeze-fracture of whole plant cells, (2) a thylakoid lumen width that is similar to previously reports from whole light-adapted algae and leaf cells, but wider that previous reports of isolated plant thylakoids.

**Recommendations for the authors:**

**Reviewer #1 (Recommendations for the authors):**
(1) Figure 1-3: It would be better if it was easier to see which part of the figure the explanation in the text refers to. For example, not only the figure number but also the color of the arrowheads could be indicated in the text. Also, it would be better to indicate which part of the figure the explanation in the text and in the figure legend refers to by adding arrows or circles on the figure images.

Thank you for this idea. We have added color references to individual objects segmented in Figs. 1 and 2. They are now indicated in the figure references in the text to facilitate the reading. In Fig. 3, we have added additional arrows (and indication in the text) to point to examples of Rubisco densities (as also requested by Reviewer #2).

(2) Figure 5: Without having read the authors' previous works on "menbranogram", the reader may have no idea why the distribution of PSI and ATPase in the non-stack region in G can be inferred from the data in Figure 5C-E. Is it possible to add an explanation, for example by adding a supplement figure?

Thank you for this suggestion. Instead of creating another methods figure and movie about membranograms, we refer readers to our earlier work (Wietrzynski et al. 2020, eLife). This fits with the Research Advance format, and eLife should clearly link to that previous paper that our current study builds upon.

**Reviewer #2 (Recommendations for the authors):**
Minor points:(1) Please add to Figures 2A or 3A arrowheads showing Rubisco complexes.

Done; we added colored arrowheads pointing to Rubisco complexes and an indication in the figure legend.

(2) "We measured a membrane thickness of 5.1 {plus minus} 0. 3 nm, a stromal gap of 3.2 {plus minus} 0. 3 nm, a luminal thickness of 10.8 {plus minus} 2.0 nm, and a total thylakoid thickness (including two membranes plus the enclosed lumen) of 21.1 {plus minus} 1.8 nm (Fig. 4) (for comparison see [1, 2, 30, 40])."Please add ref: Kirchhoff, H. et al. Dynamic control of protein diffusion within the granal thylakoid lumen. Proc. Natl Acad. Sci. USA 108, 20248-20253 (2011).

Thank you for this suggestion. The reference has been added.

(3) Please add to the supplemental figures a raw data and a processed image with AI denoising.

Denoising results differ between the tomograms. Below we provide an example of a significant improvement in signal to noise ratio in a denoised tomogram. On the left is a raw tomogram reconstructed using a standard approach: weighted back projection using etomo program from the IMOD package. On the right is the same tomogram denoised using cryoCARE, which performs a noise comparison between odd and even frames that were used to reconstruct the tomogram on the left. Below is a zoom in into the slices from the first row, highlighting the differences. The same approach was used for all the tomograms used in the figures. Please also see the Data deposition statement below (and the Data deposition section in the paper) that we hope fulfills the Reviewers request. All raw and denoised data, as well as segmentations and picked particle positions, are publicly available.

“Data deposition statement

The raw data consists of micrographs (frames) used to reconstruct each tomogram, acquisition parameters file (.mdoc) for each tomogram and reference images of the microscope camera: 273.7 GB in total. Following the current standard in the cryo-EM field, all images used to generate figures in the manuscript (AI-denoised tomograms and corresponding segmentations) have been deposited in the Electron Microscopy Data Base (EMDB) and are available under accession codes EMD-5243 through EMD-5248. They can be accessed here: https://www.ebi.ac.uk/emdb/EMD-52542. Additionally, all raw files (including tomograms used only for analysis), all used denoised tomographic volumes and unaltered membrane segmentations have been deposited onto the public EMPIAR server (https://www.ebi.ac.uk/empiar/) and are available under the accession code EMPIAR-12612. Finally, positions of PSII particles used in the study, segmented single membrane instances and membrane meshes are available at: 10.5281/zenodo.15090119. All this data will be linked to (and is searchable by) the EMDB depositions and to manuscript DOI. Accession numbers to the data are added in the “Data availability” section of the manuscript.”

**Author response image 1. sa3fig1:** Results of tomogram denoising. An example tomogram from the dataset. Top row: on the left is a 5-slice average of the tomographic volume reconstructed using weighted back projection method. On the right is a single tomographic slice of the same tomographic volume denoised using cryoCARE program. Bottom row: zoom-ins into the corresponding tomographic slices from the top row. All images were recorded using 3dmod from the IMOD package.

Additional modifications:

Following other comments and suggestions, we have included following additions to the manuscript:

Figure 4 – figure supplement 1. Its aim is to better explain the methodology behind thylakoid width measurements. The methods section concerning this figure has been slightly modify to match this addition.

Figure 1 – video supplement 1. Overview of a chloroplast tomogram and segmentations the thylakoid and chloroplast envelope membranes.

Figure 3 – video supplement 1. Chloroplast stroma and top views of the thylakoid network, with stromal lamellae connecting the grana.

Figure 8 – video supplements 1 and 2. These tomographic views highlight the organization of PSII particles in thylakoids from intact and broken chloroplasts.